# Sensitivity towards elevated $p$CO$_2$ in great scallop (*Pecten maximus* Lamarck) embryos and fed larvae.

Sissel Andersen[1], Ellen S. Grefsrud[2], Torstein Harboe[1]

[1] Institute of Marine Research, Austevoll Research Station, N-5392 Storebø, Norway
[2] Institute of Marine Research, Postbox 1870 Nordnes, N-5817 Bergen, Norway

*Correspondence to*: Sissel Andersen (sissel.andersen@imr.no)

**Abstract.** The increasing amount of dissolved anthropogenic CO$_2$ has caused a drop in pH-values in the open ocean known as ocean acidification. This change in seawater carbonate chemistry has been shown to have a negative effect on a number of marine organisms. Early life stages are the most vulnerable, and especially the organisms that produce calcified structures in the phylum Mollusca. Few studies have looked at effects on scallops, and this is the first study presented including fed larvae of the great scallop (*Pecten maximus*) followed until day 14 post-fertilization. Fertilized eggs from unexposed parents were exposed to three levels of $p$CO$_2$ using four replicate units: 465 (ambient), 768 and 1294 µatm, corresponding to pH$_{NIST}$ of 7.94, 7.75 (-0.19 units) and 7.54 (-0.40 units), respectively. All of the observed parameters were negatively affected by elevated $p$CO$_2$: survival, larval development, shell growth and normal shell development. The latter was observed to be affected only two days after fertilization. Negative effects on the fed larvae at day 7 were similar to what was shown earlier for unfed *P. maximus* larvae. Growth rate in the group at 768 µatm seemed to decline after day 7, indicating that the ability to overcome the environmental change at moderately elevated $p$CO$_2$ was lost over time. The present study shows that food availability does not decrease the sensitivity to elevated $p$CO$_2$ in *P. maximus* larvae. Unless genetic adaptation and acclimatization counteract the negative effects of long term elevated $p$CO$_2$, recruitment in populations of *P. maximus* will most likely be negatively affected by the projected drop of 0.06 – 0.32 units in pH within year 2100.

**Keywords**: Ocean acidification; bivalve larvae; scallop; *Pecten maximus*; deformity; elevated $p$CO$_2$

## 1. Introduction

The Intergovernmental Panel on Climate Change (IPCC) affirms that the uptake of anthropogenic CO$_2$ in the ocean has very likely caused elevated seawater CO$_2$ levels and thereby lowered the average oceanic pH values, termed ocean acidification (IPCC, 2013). A great effort is initiated worldwide to increase our knowledge of how ocean acidification affect coastal marine organisms; producing growing evidence that a high number of species respond negatively to exposure to elevated CO$_2$ levels (Kroeker et al., 2013). It is crucial to gain more knowledge about the effects on a range of marine organisms in order to get realistic projections of future changes to the marine ecosystems.

Calcifying organisms seem to be more sensitive to elevated $CO_2$ than non-calcifying organisms, and early life stages are more sensitive than older individuals (Byrne, 2012). Studies on bivalves, especially mussels and oysters, have reported negative effects on the pelagic early life stages (Gazeau et al., 2013; Kroeker et al., 2013; Parker et al., 2013). However, sensitivity to elevated $CO_2$ is species specific and can vary greatly in closely related species, between populations and even between individuals (Arnold et al., 2009; Ries et al., 2009; Parker et al., 2011; Agnalt et al., 2013). The great scallop, *Pecten maximus*, is a commercially exploited species in several European countries (Brand, 2006; Norman et al., 2006; Strand and Parsons, 2006), found in coastal shell sand areas mainly at depths of 10-50 meters, making this species highly exposed to changes in the coastal environment throughout its life cycle from the planktonic larvae to the benthic adults. To our knowledge only four studies have been published on the effects of elevated $CO_2$ on the great scallop, one on unfed larvae (Andesern et al., 2013a), one on juveniles (Sanders et al., 2013) and two on adults (Schalkhausser et al., 2012, 2014). The studies of larvae and adults showed a negative effect of elevated $CO_2$ on the scallop, while the experiment using juveniles concluded that *P. maximus* is potentially tolerant to elevated $CO_2$ when food is unlimited. Blue mussel (*Mytilus edulis*) juveniles also seem to manage elevated $CO_2$ better at higher food concentrations (Thomsen et al., 2010, 2013; Melzner et al., 2011). Thomsen et al. (2010, 2013) showed that food availability outweighs acidification effects in juvenile *M. edulis* and Melzner et al. (2011) showed that low food levels gave a negative effect on internal shell dissolution independent of $p$$CO_2$ levels in the same species. The importance of the energy budget under conditions of $CO_2$-stress was also shown in unfed six- and eight-day-old sea urchin larvae (*Strongylocentrotus purpuratus*) when they lowered their ATP allocation to protein synthesis and in vivo Na+, K+-ATPase activity compared with fed larvae (Pan et al., 2015).

Andersen et al. (2013a) showed that survival and growth in larvae decreased when $CO_2$ exposure started with fertilized eggs. In addition, the percentage of deformed larvae increased. This study covered the first seven days of embryonic and larval development in unfed larvae. Since energy levels in larvae are critical for normal growth and development (Delaunay et al., 1992; Nevejan et al., 2003), the question was raised whether lack of food may have added stress to the larvae, making them more vulnerable to the elevated $p$$CO_2$ levels. In the present study we reared scallop larvae throughout a 14-day period with a feeding regime normally used in aquaculture production (Andersen et al., 2011, 2013b) aiming to elucidate whether larvae offered food would be more resilient to elevated $CO_2$.

## 2. Materials and Methods

Local broodstock of the great scallop *Pecten maximus* were collected in January 2013 from Hardangerfjorden on the west coast of Norway. They were transported to the experimental hatchery at the Institute of Marine Research (IMR) - Austevoll Research Station, cleaned of fouling and deployed in running seawater at an initial temperature of 8.8 ℃. The broodstock was conditioned and spawned according to standard procedures at IMR (Andersen et al., 2013; Andersen et al., 2011) using the same seawater quality (ambient) as was later offered to the larval control group. Spawning was induced by a temperature increase on March 6. *P. maximus* is a simultaneous hermaphrodite, and only egg batches with less than 10 % self-fertilization were used in the experiment. The final fertilization (cross and self) was around 87 %.

Fertilized eggs were incubated at a density of 13 eggs mL$^{-1}$ in 38 L exposure tanks at ambient pH$_{NIST}$ 7.94 (control) and mean pH$_{NIST}$ of 7.75 and 7.54, corresponding to a $p$CO$_2$ of 465, 768 and 1294 µatm, respectively (table 1). Four replicate tanks were used per pH-treatment. The pH levels were chosen based on the predicted drop of 0.5 units in the open ocean from today to year 2250 (IPCC 2013), and kept within the range used by Andersen et al., (2013a). IPCC (2013) has projected the pH levels by 2100 to be 0.06 to 0.32 lower than it is today.

The sea water supply and experimental design are described in detail in Andersen et al. (2013a), and an overview is shown in figure 1. The mesh sizes of the outlet sieves in the centre of the larval tanks were 35, 41 and 63 µm at days 0-2, 2-7 and 7-14, respectively. The measured water flow at day 1 was 6.6 ± 0.6 L hour$^{-1}$ (mean ± SD, n=12), and at days 2, 7 and 8: 11.5 ± 0.7 L h$^{-1}$ (mean ± SD, n=36) corresponding to an exchange rate of 7.3 times day$^{-1}$.

## 2.1. Seawater parameters

Seawater was pumped from 160 m depth and filtered through a sand filter before temperature was adjusted in a heat pump. The water was aerated and finally filtered through a 50 µm filter. Temperature was recorded every 10 minutes using a four detector (one in air and three in exposure tanks) EBI – 1 Ebro 4 temperature logger. The overall mean temperature (± SD) calculated from recordings every 10 minutes in three tanks (at the three treatments), was 15.48 ± 0.16 $^{\circ}$C (n=3903). Daily means based on recordings every 10 minutes for each treatment (table 1) was used to calculate $p$CO$_2$ values. Salinity was checked daily using a WTW LF330 Conductivity meter.

The pH-level in each exposure tank was measured daily in a 100 mL sample using a Mettler Toledo equipped with a Metler Toledo InLab®ExpertPro pH-probe, calibrated with 4.00 and 7.00 buffers (Certipur® buffer solutions, Merck KGaA, 64271 Damstadt, Germany) traceable to standard reference material from NIST (NBS). The daily means for each treatment (table 1) was used to calculate $p$CO$_2$ values.

Total alkalinity (A$_T$) was analyzed in the three treatments at the start and end of experiment (n=6) by a Titralab, Radiometer, and the mean value 2321.5 µmol kgSW$^{-1}$ was used when calculating $p$CO$_2$ values.

The $p$CO$_2$-values (µatm) corresponding to the pH$_{NIST}$-values (table 1) were calculated based on the means of temperature ($^{\circ}$C), pH$_{NIST}$, salinity and A$_T$, and using the macro taken directly from Ernie Lewis' "CO2SYS.BAS" Basic Program (Pierrot et al., 2006) with the set of constants K1, K2 from Mehrbach et al. (1973) refit by Dickson and Millero (1987), the constant for KHSO$_4$ from Dickson and Millero (1987) and for total Boron (B$_T$) from Uppstrom (1974). Seawater at different pH-levels was produced by mixing seawater with an acid stock solution of pH$_{NIST}$ 5.80, made from mixing CO$_2$ gas and seawater with an ambient pH$_{NIST}$ of 7.95. The pH in each mixing tank was continuously adjusted to pre-set levels by addition of stock solution with dosage pumps (IWAKI) controlled by feedback from pH-electrodes to pH-transmitters (Endress & Hauser).

## 2.1. Larval diet

The larval diet was a standard mixture of three algae species (Mackie et al., 1984; Andersen et al., 2011, 2013b): *Isochrysis galbana* (Tahitian strain), *Pavlova lutherii* and *Chaetoceros mulleri*. Algal concentration and particle sizes were monitored using an electronic particle counter, Coulter Z2 (Beckman Coulter). Mean cell volumes (± SD) of the three species during the experiment were $44.7 \pm 4.2$, $45.9 \pm 7.6$ and $68.2 \pm 8.4$ µm$^3$ (n=12), respectively. The theoretical algal concentration fed into the inlet seawater was 3 cells µl$^{-1}$ at days 2-4, 6 cells µl$^{-1}$ at days 5-6, 10 cells µl$^{-1}$ at days 7-10 and 15 cells µl$^{-1}$ at days 11-14. The diet was pumped to the tanks in 15:15 minute pulses (on:off) over a period of 20-22 hours.

Measurements showed that mean algal cell concentration in the water running out of the tanks was 8 and 6 cells µl$^{-1}$ at days 8 and 10 post-spawn, respectively (table 2). Mean concentration inside the tanks was 9 and 10 cells µl$^{-1}$ at days 10 and 11, respectively.

## 2.2. Sampling of larvae

Larvae were sampled by collecting 700 mL from each replicate at days 2, 3 and 7. At day 14 all tanks were drained and a total sample from each replicate was collected. Sampling and preservation in 4 % formalin is described in detail in Andersen et al. (2013a). Larvae in the samples were preserved to measure shell size, survival and to determine if the larval shell was normally developed or deformed.

To calculate the survival based on the initial number of fertilized eggs, larvae concentrated in smaller volumes were counted in 4 x 200 µl droplets at days 3 and 7, and in 10 x 50 µl droplets at day 14. To make our results comparable with the literature on scallop spat-production (Andersen et al., 2011; Magnesen et al., 2006) we estimated survival at days 7 and 14 also based on the day 3 yield.

Shell length (parallel to the hinge) and shape was investigated using photos, as described in Andersen et al. (2013). The number of individuals classified as "live" that was measured from each replicate was 74-104 at day 3, 67-97 at day 7 and 92-156 at day 14.

At day 2 larvae that had not yet developed visible shell valves were classified as "unshelled" larvae. At day 3 larvae that had not developed the muscle to retract the velum was identified, after being preserved, by the presence of a protruded velum. They were classified as larvae with a "protruded velum". These larvae were not considered to be abnormal and the protruded velum not an artifact of preservation, but they were considered to be larvae that had developed slower than shelled larvae or larvae without a protruded velum, respectively.

Larvae at days 3, 7 and 14 were classified in four categories according to shell shape (Andersen et al., 2013a): 1) *Normal*, 2) *Hinge* deformity, 3) *Edge* deformity, and 4) *Both* (edge and hinge deformity). Trochophore larvae at day 2 were only classified in category 1 and 2 (*Normal* and *Hinge* deformity), since the shell edge was often hidden by soft tissue. The number of live larvae classified per replicate (independent of treatment) at day 2 was 96-150, and at days 3, 7 and 14 it was similar to the number of live individuals used for shell length measurements. Larvae were classified as "live" when the shell

was filled with soft tissue, and as "dead" when the shell was empty or contained little soft tissue. Deformities in dead larvae were classified only at day 14 based on 29-181 dead individuals (out of 122-333 individuals) from the different replicates, since there were too few dead individuals in the larval samples at days 3 and 7. To describe the relative variation in shell shape categories between replicates for the three treatments, the coefficient of variation (CoV) was calculated as percentage sd of means:

CoV=sd x 100/mean

### 2.3 Statistics

To find effects of $pCO_2$ the parametrical tests one-way ANOVA (ANOVA) and GLM followed by Tukey's HSD post-hoc test to find differences between groups, were used if the data or transformed data conformed to normality using Shapiro-Wilk´s W test and the variances were homogeneous according to Levene's test. Effect of days on *Normal* shell category was tested using GLM. A t-test was used to determine if there were differences in shell shape (normal or deformed) between live and dead larvae. Results given as percentages (survival and shell shape categories) were arcsine transformed prior to testing. When parametrical tests were inappropriate, Kruskal-Wallis ANOVA by ranks (K-W ANOVA) was used to test effects of $pCO_2$, and differences between groups were then tested using p-values for Multiple Comparisons (2-tailed). To find if normally developed larvae at day 7 post-spawn were different from day 14, the Kolmogorov-Smirnov Test was used for the two elevated $pCO_2$ groups. To find differences between groups, a non-parametric t-test, the Mann-Whitney U test, was used when Multiple Comparisons did not show differences between groups even if the Kolmogorov-Smirnov Test showed significant effects. The significance level used in all tests was set to 0.05. Statistica version 11 (Statsoft Inc.) was used to run all statistical tests.

## 3. Results

### 3.1. Survival

The mean survival at day 3 post-spawn based on the initial egg count varied between 27.6 % and 31.1 % for the three $pCO_2$ groups (Fig. 2). After day 3 survival decreased more at 1294 µatm than at lower $pCO_2$ and was only 3.1 % by day 14, while it was 18.2 and 14.9 % in the ambient group ( 465 µatm) and at 768 µatm, respectively.

Mean survival was lower at 768 µatm than in the ambient group (465 µatm) at all days, but the differences were not significant. Significant differences in survival between $pCO_2$ groups were only found between the highest $pCO_2$ and the other two groups at day 7 (p=0.001 for 465 and 1294 µatm; p= 0.028 for 768 and 1294 µatm) and day 14 (p<0.001 for all). Based on survival at day 3 the present study gives a survival at day 14 of 62, 57 and 10 % for the groups at 465, 768 and 1294 µatm.

## 3.2. Shell development and length

Larvae that had not yet developed a visible shell (unshelled larvae) at day 2 ranged from 4.0 % in the ambient group to 9.8 % and 23.4 % in the 768 and 1294 µatm groups, respectively (Fig. 3). A significant difference in unshelled larvae was only found between the lowest and highest $pCO_2$ (p=0.032). At day 3 no unshelled larvae were observed, but larvae with a protruded velum were observed in all groups (Fig. 3). The percentage of larvae with a protruded velum was affected by $pCO_2$ (p=0.025) and increased from 4.6 and 4.5 % in the ambient and 768 µatm group, respectively, to 33.7 % at 1294 µatm. Larvae with a protruded velum were also found at day 7, but in only 3 out of a total of 1134 individuals, and were not observed at day 14.

At day 3, shell length (SL) of larvae was 109.9, 107.2 and 94.6 µm at 465 µatm (ambient), 768 µatm and 1294 µatm, respectively (Fig. 4). However, shell growth at 768 µatm was slower than in the ambient group and the difference in SL between the two increased until day 14 when the 768 µatm group was similar in shell length to the 1294 µatm group (Fig. 4), with average values of 123.7 and 122.7 µm, respectively. At day 14 SL of the ambient group was 134.4 µm.

There was an effect of $pCO_2$ at all days post-spawn (day 3: p<0,001; day 7: p=0.002; day 14: p<0.001). All $pCO_2$ groups were significantly different in SL at day 3 (p<0.013 for all groups), at day 7 only SL at the highest $pCO_2$ was significantly different from the ambient group (p=0.002), but at day 14 SL of both elevated groups were significantly different from the ambient group (p<0.001).

The mean shell growth rate for the days 3-14 was 2.2, 1.5 and 2.5 µm day$^{-1}$ in the $pCO_2$ groups 465, 768 and 1294 µatm, respectively, and it was significantly lower in the 768 µatm group than in the other two groups (p<0.004 for both groups). Larvae in the two elevated $pCO_2$ groups showed different shell growth patterns, with a higher daily growth rate at the most elevated level, but starting from a smaller SL at day 3 (Fig. 4).

## 3.4. Shell shape categories

On day 2 90, 89 and 65 % of the live larvae at ambient, 768 and 1294 µatm, respectively, had a normal hinge, and the most elevated group was significantly different from the other two (p<0.001 for both). The percentage of live larvae classified in the category *Normal* at day 3-14 decreased when $pCO_2$ increased independent of days, and the values were lowest at day 7 in all groups (Fig. 5A-C), but there was no significant effect of days on *Normal* in any of the treatments. The range of *Normal* in the ambient group (437 µatm) for day 3-14 was 71-80 %, and the effect of $pCO_2$ was significant on all days (p<0.001 for day 3; p=0.007 for both day 7 and day 14). Only the most elevated group was significantly different from the ambient at all days (p<0.001 for day 3; p=0.005 for both day 7 and day 14), except at day 3 when all groups were significantly different (p=0.043 for 465 and 768 µatm, p<0.001 for both 465 and 768 µatm, and 1294 µatm).

*Edge* was the most frequently occurring deformity category at day 3-14, observed in 15-25 % of the ambient larvae and in 32-63 % of the elevated groups (Fig. 5A-C). The *Hinge* category was always highest in the most elevated $CO_2$-group with a range of 5-30 % at day 3-14 (Fig. 5A-C), and the values were mostly more than double the values in the other two groups

(range 2-6 %). Few larvae were classified in the category *Both*, but the ranges increased with an increase in $pCO_2$ level: 0.3-1.5 %, 1.1-7.0 % and 9.4-31.7 % at 465, 768 and 1294 µtam, respectively (Fig. 5A-C).

The variation between replicates in the different shell categories was relatively high in the two groups at elevated $CO_2$. The Coefficient of Variation (CoV) ranged between 2.4 % at day 2 (at 768 µatm) and 115.5 % at day 7 (at 1294 µatm). For the three deformity categories *Edge, Hinge* and *Both* the CoV range was 18.9-146.4 %. In general, the variation was highest at day 7. In two out of four replicates in the most elevated group at day 7 we did not observe any live larvae in the category *Normal* (n=94 and 83), but in the same replicates there were normally shaped dead larvae (6.3 and 20.0 %).

In dead larvae at day 14 *Normal* decreased from 55 % at 465 µatm (ambient) to 25 % at 1294 µatm (Fig. 5D). There was an effect of $pCO_2$ on *Normal* (p=0.018), and 465 µatm (ambient) was significantly different from 1294 µatm (p=0.018). *Edge* was also the major category observed in dead larvae, and ranged from 38 % in the ambient group to 54 % at 1294 µatm (Fig. 5D). Shell deformity in dead larvae was significantly higher than in live larvae at day 14 (Fig. 5C and D) in the ambient (p=0.011) and the 768group (p<0.001), but not in the most elevated group.

## 4. Discussion

The present study shows the effects of elevated $pCO_2$ on *Pecten maximus* embryos and fed larvae during a 14-day period, approximately two thirds of the larval lifecycle. Although larvae may experience lack or scarcity of food in their natural environment it is important to know if sufficient food supply decreases larval sensitivity towards elevated $pCO_2$ as the early life stages of bivalves seem to be very sensitive to elevated levels of $CO_2$ (Fabry et al., 2008; Kurihara 2008; Talmage and Gobler 2009, 2010, 2011; Parker et al., 2010; Beniash et al. 2010; Gazeau et al., 2010; Gaylord et al., 2011; Andersen et al., 2013a; White et al., 2013, 2014). A number of studies have shown effects on marine invertebrate larvae (Brennand et al., 2010; Crim et al., 2011; Stumpp et al., 2012), but only Andersen et al. (2013a) have studied *P. maximus* embryos and larvae. Andersen et al. (2013a) presented a comparison of studies concluding that the responses to elevated $pCO_2$ seem to vary little between bivalve species, but the magnitude of the responses may differ. In addition, the $pCO_2$ level, temperature and rearing volume in the studies vary, and one should therefore be careful in drawing conclusions about effects between studies.

### 4.1. Food availability

Scallop larvae were fed about 6-10 cells µl$^{-1}$, which was within the standard range described by Magnesen et al., (2006) and Andersen et al., (2013b). Andersen et al. (2013b) showed that feed concentrations of 3-20 cells µL$^{-1}$ in large rearing volumes (2800 L) affected the lipid content in the larval populations but not larval survival or total yield of juveniles four weeks after metamorphosis. Based on these results feed concentrations in the present study should be sufficient to maintain growth and survival at this larval concentration and age.

## 4.2. Survival

In the present study survival in the ambient and 768 µatm groups was not significantly different throughout the experimental period, while survival in the 1294 µatm group was lower than in the ambient group at day 7 and day 14. When $pCO_2$ increased from 465 (ambient) to 1294 µatm the survival decreased with a factor of 0.3 on day 7 and 0.2 on day 14, similar to what was observed in Andersen at al. (2013a). The negative effect of elevated $pCO_2$ on survival relative to the ambient group was similar in both studies, thus we could not detect any positive effect of feeding. Mean survival at day 7 in the ambient group was 21 % of the initial egg count, less than half of that reported by Andersen et al. (2013a) (45 %) using the same rearing system without food supply. The differences in survival of the ambient groups between the two studies may have been caused by slight changes in experimental conditions or by parental effects between larval groups (Andersen et al., 2011).

The incubated eggs yielded 28-31 % larvae collected on a 35 µm mesh screen at day 3 post-spawn, independent of $pCO_2$ treatment. The survival until day 3 in our study was on the lower side of what was earlier reported for larger rearing systems (Andersen and Ringvold 2000; Andersen et al., 2000; Magnesen et al., 2006), and may be due to the smaller rearing volumes or other unknown factors. Survival of more than 60 % at day 14 in the ambient group, based on the larval count at day 3, was in accordance with survival in batches of viable *P. maximus* larvae in large scale hatchery rearing systems (Andersen et al., 2011, 2013b). This indicates that the veligers in our study were viable and healthy. The survival at day 14 based on day 3 fits well for the two lowest $pCO_2$-groups with the relationship between growth rate and survival that was shown in Marshall et al. (2010), but survival at the highest $pCO_2$ was much lower than they described.

## 4.3. Larvae development and shell size

At the earliest shelled stage, the muscles are not sufficiently developed and thus the larvae are unable to retract the velum (Cragg 2006 and references therein). The percentages of unshelled larvae at day 2 and larvae with a protruded velum at day 3 were significantly higher in the 1294 µatm group, which is most likely a result of delayed development caused by elevated $pCO_2$, is in accordance with the reports of slower development at elevated $pCO_2$ levels reported in the earlier study of great scallop larvae (Andersen et al., 2013a) and also in other bivalve larvae (Talmage and Gobler 2011; Kurihara 2008).

Larvae in the ambient group and at 768 µatm were larger than larvae at 1294 µatm at days 3 and 7. At day 14, larvae at 768 µatm were smaller than in the ambient group, but similar in size to the larvae in the 1294 group. The change in growth rate after day 7 as seen in the 768 µatm group has not been shown for scallop larvae earlier, as Andersen et al. (2013a) ended their study at day 7. This indicates that the growth rate in larvae in the 768 µatm group may have been able to compensate to some extent for the elevated $pCO_2$ level for the first seven days. Later the growth rate was affected by the carbonate chemistry, even if survival seemed unaffected. It may then be discussed if the endogen reserves for the current larval groups were insufficient for embryo and first stage larvae development at the highest $pCO_2$ level, since survival decreased and the remaining larvae developed slower compared to the two lower $pCO_2$ groups. Andersen et al. (2013b) showed that larvae

supplied with no food after day 3, stopped growing at day 6 while the survival was affected only after day 15. This indicated that endogen reserves from the eggs (Cragg, 2006) were important for larval growth and development until day 6, and that exogenous energy from food would be more important after that day. The change in growth rate after day 7 for the 768 µatm group also shows the importance of longer exposure duration to find indications of future success for larval groups. Larvae in the 1294 µatm group developed slower than the other two groups until day 3, and the smaller size was uncompensated until day 14 when the experiment ended. This was also shown by White et al. (2014) when they exposed one group of Bay scallop larvae, *Argopecten irradians*, to elevated $pCO_2$ from 11 hours post-fertilization and for only three days. The negative effect on size was not compensated for when larvae were transferred back to ambient conditions, although they were pulse fed daily. Our larvae in the 1294 µatm group reach the same size at day 14 as larvae in the ambient group reached at day 7.

Shell length for the two lowest $pCO_2$ groups in our study were only slightly higher than reported by Andersen et al. (2013a) at day 7 (119 µm in our ambient group vs 115 µm in theirs, and 115 µm in our 768 µatm group vs 110 µm in their 821 µatm group), and shell length was the same for the most elevated groups (105 µm). The similar shell length at day 7 for the two larval batches suggests that their shell growth rate was similar. Andersen et al. (2013) showed that food availability did not affect larval shell growth the first 6 days after fertilization, supporting that feeding probably did not cause any difference in growth rate between the two larval batches.

**4.4. Shell deformities**

The effect of $pCO_2$ on shell shape of day 2 larvae seemed to be less in our study compared with Andersen et al. (2013a). The range of normally developed hinges in the present study was 65-90 % while Andersen et al. (2013a) reported 28-68 %. Since food was distributed after day 2 this was not due to feeding, but may have been caused by other factors such as genetic variation or energy status. It is known that the variation in performance between larval batches of *P. maximus* is very high (Andersen et al., 2011), and a variation in egg quality have been suggested as an explanation by Robert and Gérard (1999).

The proportion of deformed live larvae in the ambient group was 20-29 % at days 3-14. This may seem high in comparison to His et al. (1997) who found that it was 5-10 % in the control groups of 18h and 48h old embryos or larvae of the oyster *C. gigas* and mussel *M. galloprovincialis*, respectively, when kept in static seawater in 25 mL containers. A relatively high deformity level in the ambient group indicates that other factors than the treatment may also cause physiological stress to the embryo and larvae. However, the deformity level in our larvae at ambient $pCO_2$ was lower than the nearly 40 % reported for unfed scallop larvae in the same rearing system at day 7 (Andersen et al., 2013). This could indicate that food counteract deformity at ambient levels of $pCO_2$ or simply be a natural difference between larval groups due to genetics or parental effects.

The different types of deformity seemed to show different patterns with an increase in $pCO_2$. Edge deformity was higher in the two elevated groups than in the ambient group, while hinge deformity was higher in the most elevated group than in the two other groups. The percentage of *Normal* followed a more linear pattern with an increase in $pCO_2$ independent on days.

The percentage of larvae in the *Edge* category in our study on day 7 was similar to the percentages reported by Andersen et al. (2013a), 25-63 % in ours vs 30-57 % in theirs. In *Hinge* our percentages were lower (2-5 % vs 5-22 %) and in *Both* our percentages were higher (1-32 % vs1-10 %). Again, differences between family groups may explain the variation found between the two experiments. The high Coefficient of Variation (CoV) also indicates that factors differing between replicates may partly contribute to deformities. However, the present study shows that more larvae develop shell deformities with elevated $pCO_2$ and that feeding does not seem to counteract this effect.

The aragonite saturation at 0.82 can possibly add energetic stress to the larvae, since calcium carbonate dissolves at saturation below 1 (Andersson et al., 2011). However, the carbonate shell in live larvae is covered by a protein layer, the periostracum (Mouëza et al., 2006; Silberfeld and Gros, 2006), and the effect of a reduced aragonite saturation may not be significant.

## 4.5. Concluding remarks and future work

Scallop embryos and larvae seem highly sensitive to elevated $pCO_2$ at a very early stage in life. Our study confirms that even when food is supplied, increased $pCO_2$ levels that may be reached within the next 50-100 years in the open ocean (Zondervan et al., 2001; IPCC 2013) have a negative effect on scallop larvae similar to that found in unfed larvae (Andersen et al., 2013a). In our study negative effects on survival were observed after seven days, size was affected after three days at the highest $pCO_2$ and development rate and normal shell development were affected after only two days at 1294 µatm. This shows that slow development rate and abnormal shell development are very early indicators of sensitivity towards stressors like elevated $pCO_2$. If this sensitivity is true also for natural populations, it could have serious implications for recruitment of natural populations and for future aquaculture production. In aquaculture, however, scallop spat is produced in land based nurseries where seawater quality can be adjusted.

Our study ended at day 14, around one week prior to metamorphosis. Future studies should focus on how well exposed larvae groups succeed through the energy-demanding process of metamorphosis, as this may be one of the main bottlenecks in the recruitment process since low survival in this stage is shown in spat production (Andersen et al., 2011). Based on earlier studies (Parker et al., 2011, 2012; Suckling et al., 2014; White et al., 2014; Jager et al., 2016;) it seems that a better understanding of energy budgets, genetic drifting and adaptation and effects of parental exposure on both gametes and larvae will be crucial to predicting recruitment success in scallop populations exposed to an increasing ocean acidification.

In most studies, $pCO_2$ levels are kept constant during the experimental period, while in the natural environment the levels fluctuate on both shorter and longer temporal scales especially in coastal areas. Fluctuations may cause extra stress to organisms exposed to a rising $pCO_2$ in the environment (Almén et al., 2014), and should be included in future experiments. Not only are the fluctuations different between open ocean and coastal areas, also average levels in coastal areas are different from open ocean levels. The few reports on the situation in near shore waters show pH values as low as 7.6, already exceeding the expected average values for the open ocean within year 2100 (e.g.; Thomsen et al. 2010; Gazeau et al. 2011; Reum et al. 2014). These data are so far based on very few coastal monitoring stations, and effort should be made to increase

the monitoring of highly productive coastal areas in the future to reveal the $p\text{CO}_2$ levels the coastal epibenthic species in fact are exposed to.

**Data availability**. The underlying research data for survival, shell length, larval development, shell shape categories and seawater chemistry can be accessed in the Supplement

**Supplement link**

**Author contribution**. All authors conceived and designed the experiments; all authors performed the experiments; Sissel Andersen contributed materials/analysis tools; Sissel Andersen and Torstein Harboe analyzed the data; Sissel Andersen and Ellen Sofie Grefsrud wrote the paper, and Torstein Harboe contributed to a lesser extent in writing.

**Acknowledgments.** This study was supported by the Institute of Marine Research through the project number 83192-04, Ocean Acidification – Scallops. We would especially like to thank Cathinka Krogness for conditioning of broodstock and Annhild Engevik for producing live algal cells of high quality. Also, we thank Dr. Anders Mangor-Jensen, Dr. Lars Helge Stien and Dr. Caroline Durif for useful discussions.

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

**Tables**

Table 1. Measured and calculated water parameters for three different $pCO_2$ groups (µatm) given as mean and standard deviation. Carbon chemistry values were computed based on daily measurements (0-14 days) of $pH_{NIST}$ in all replicates (n=4), means of hourly temperature measurements in three tanks (n=336), mean salinity and total alkalinity based on two analyses per treatment at the start and the end date, (n=6) in seawater running into the lab.

\* All measured values were the same; \*\* One mean was used for all groups

| $pCO_2$-group | 465 µatm | 768 µatm | 1294 µatm |
|---|---|---|---|
| Measured parameters | | | |
| $pH_{NIST}$ | $7.94 \pm 0.01$ | $7.75 \pm 0.01$ | $7.54 \pm 0.02$ |
| Difference from ambient | - | -0.19 | -0.40 |
| Salinity* | 35.1 | 35.1 | 35.1 |
| Temperature ($^o$C) | $15.5 \pm 0.1$ | $15.5 \pm 0.1$ | $15.5 \pm 0.1$ |
| $A_T$ (mmol/kgSW)** | $2321.5 + 4.1$ | $2321.5 + 4.1$ | $2321.5 + 4.1$ |
| Calculated parameters | | | |
| $pCO_2$ (µatm) | $465 \pm 10$ | $768 \pm 24$ | $1294 \pm 48$ |
| $HCO_3^-$ (µmol/kgSW) | $1964 \pm 6$ | $2079 \pm 6$ | $2166 \pm 6$ |
| $CO_3^{2-}$ (µmol/kgSW) | $143.7 \pm 2.3$ | $97.5 \pm 2.5$ | $62.8 \pm 2.2$ |
| $CO_2$ (µmol/kgSW) | $17.1 \pm 0.3$ | $28.2 \pm 0.9$ | $47.6 \pm 1.9$ |
| $\Omega_{aragonite}$ | $1.88 \pm 0.03$ | $1.28 \pm 0.03$ | $0.82 \pm 0.03$ |
| $CO_2$ (ppm) | $473 \pm 10$ | $781 \pm 24$ | $1316 \pm 53$ |

Table 2. Larval food concentration sampled from inside the tank and at the outlet. Values are means and standard deviation (n=12).

| Days post-spawn | Concentration (cells $\mu l^{-1}$) | | Biomass ($\mu m^3\ \mu l^{-1}$) | |
|---|---|---|---|---|
| | inside | outlet | inside | outlet |
| day 8 | - | 7.7 ± 1.4 | - | 341 ± 77 |
| day 10 | 9.6 ± 1.2 | 5.9 ± 0.6 | 437 ± 55 | 219 ± 28 |
| day 11 | 8.6 ± 0.9 | - | 381 ± 40 | - |

**Figure captions**

Figure 1. An overview of the experimental design showing both the mixing room for production of seawater with different levels of $pCO_2$ and the exposure room with fiberglass tanks for larvae.

5   Figure 2. Mean survival (%) based on the number of incubated eggs, and standard deviation (n=4) for larvae at three $pCO_2$ levels, 465 (ambient), 768 and 1294 µatm, at days 3, 7 and 14 post-spawn.

Figure 3. The median percentage of unshelled larvae at day 2 post-spawn, and larvae with protruded velum at day 3 post-spawn, upper and lower quartile (n=4) at three $pCO_2$ levels, 465 (ambient), 768 and 1294 µatm.

Figure 4. Mean shell length (µm) and standard deviation (n=4) for larvae at three $pCO_2$ levels, 465 (ambient), 768 and 1294 µatm, at days 3, 7 and 14 post-spawn.

Figure 5. Larvae classified in four different shell categories: Normal, Edge (deformities), Hinge (deformities) and Both (edge
15   and hinge deformities) at three $pCO_2$ levels, 465 (ambient), 768 and 1294 µatm, in live larvae at day 3, day 7, day 14, and in dead larvae at day 14. The figure is based on the sum of 4 replicates in each $pCO_2$ group.

**Figure 1. Experimental design.**

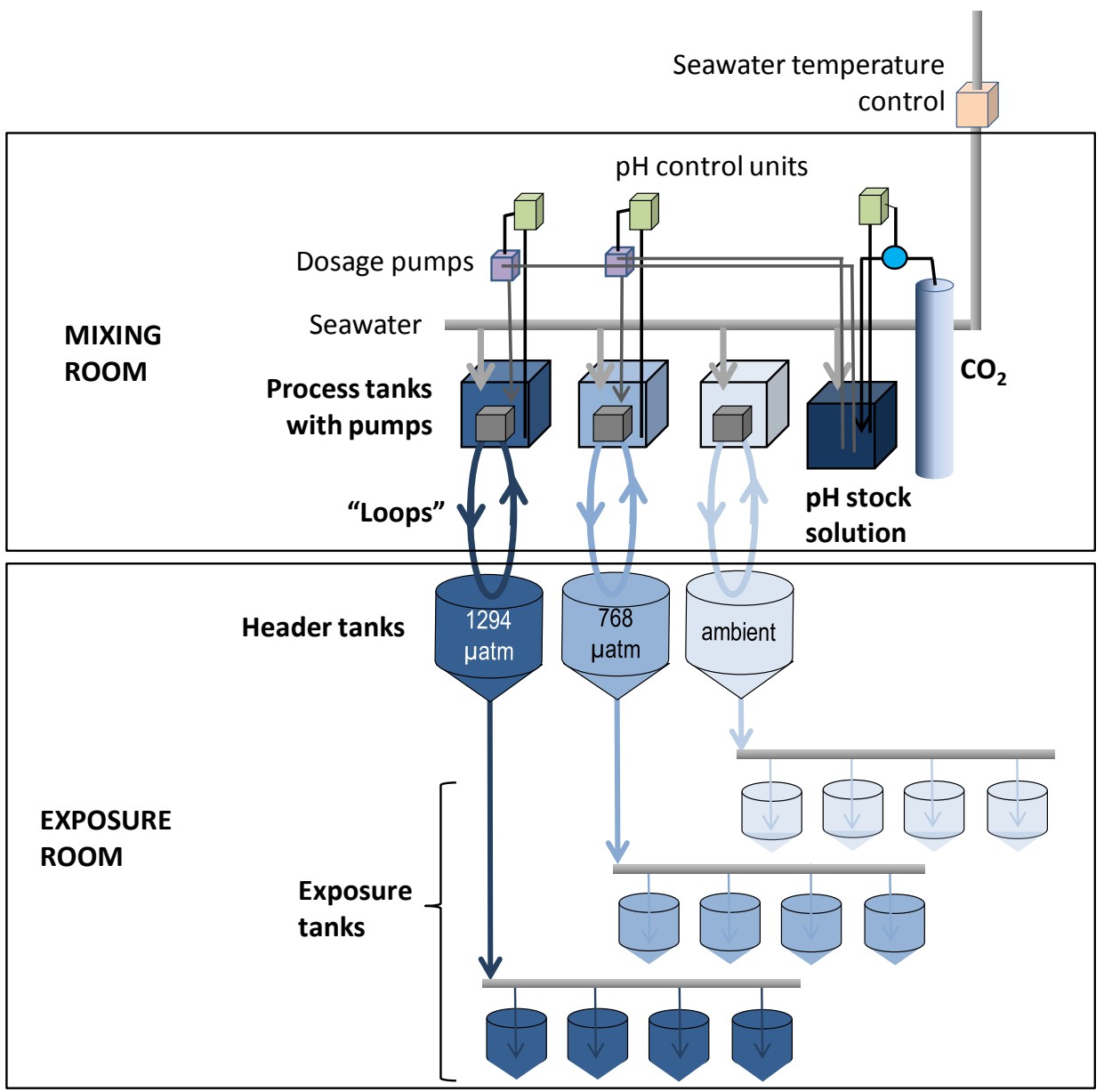

**Figure 2: Survival.**

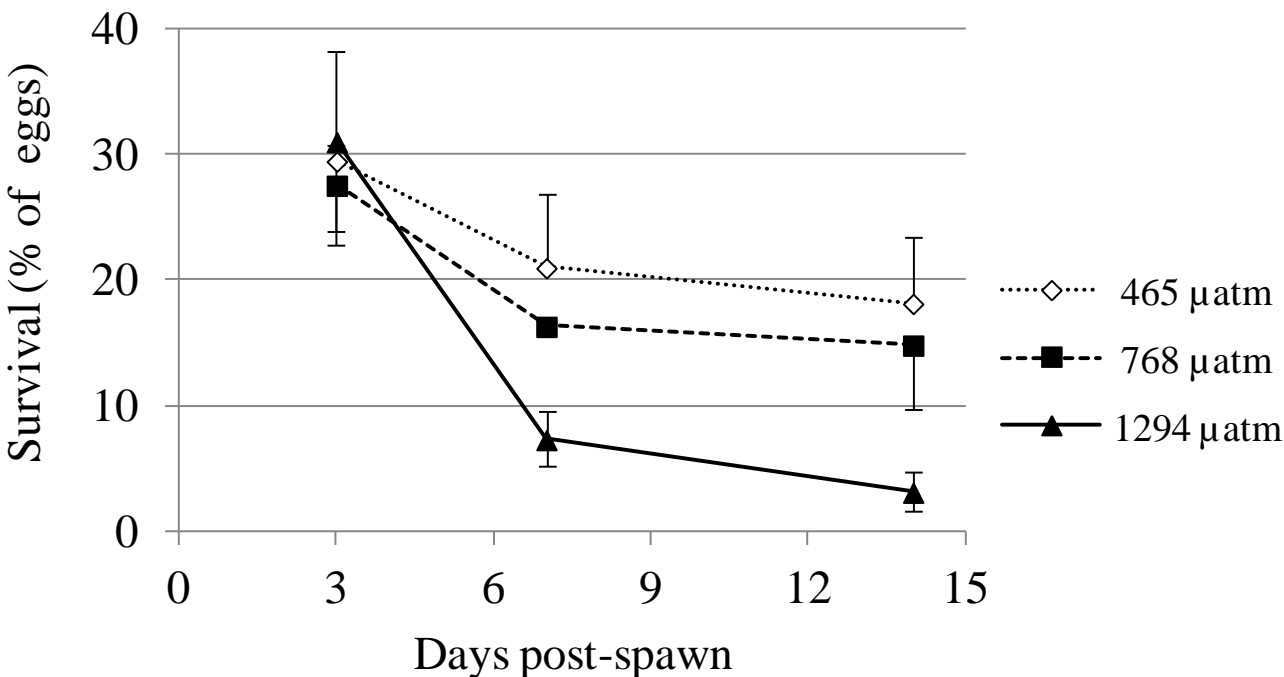

**Figure 3: Larval development.**

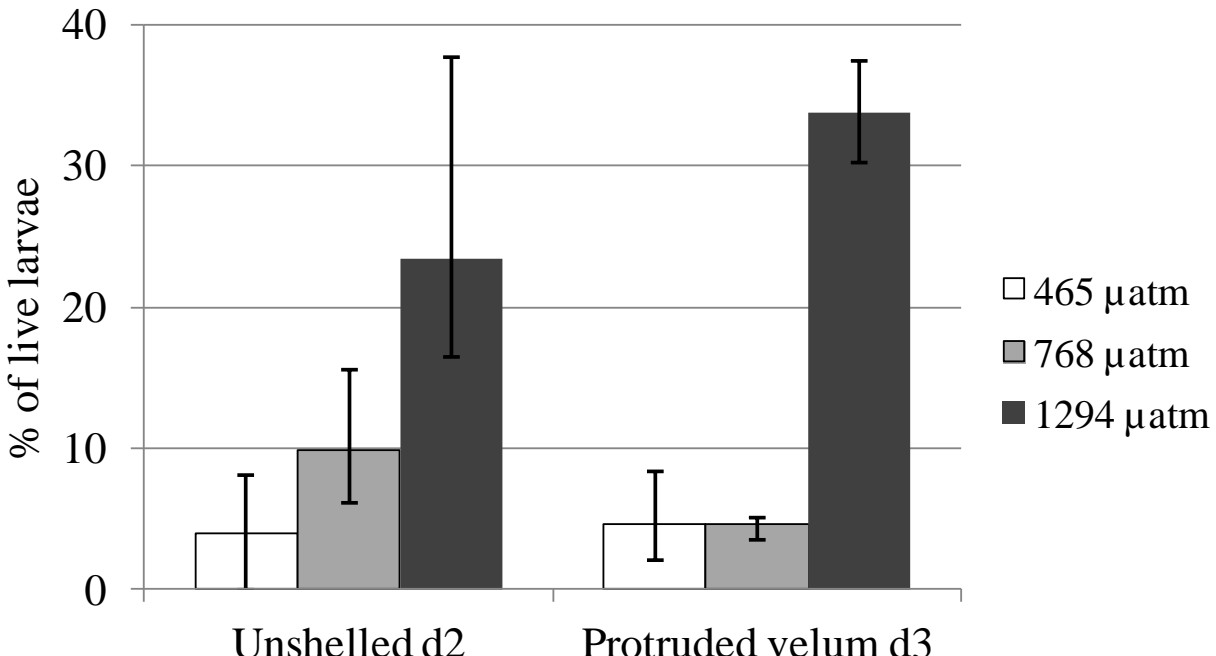

**Figure 4: Shell length.**

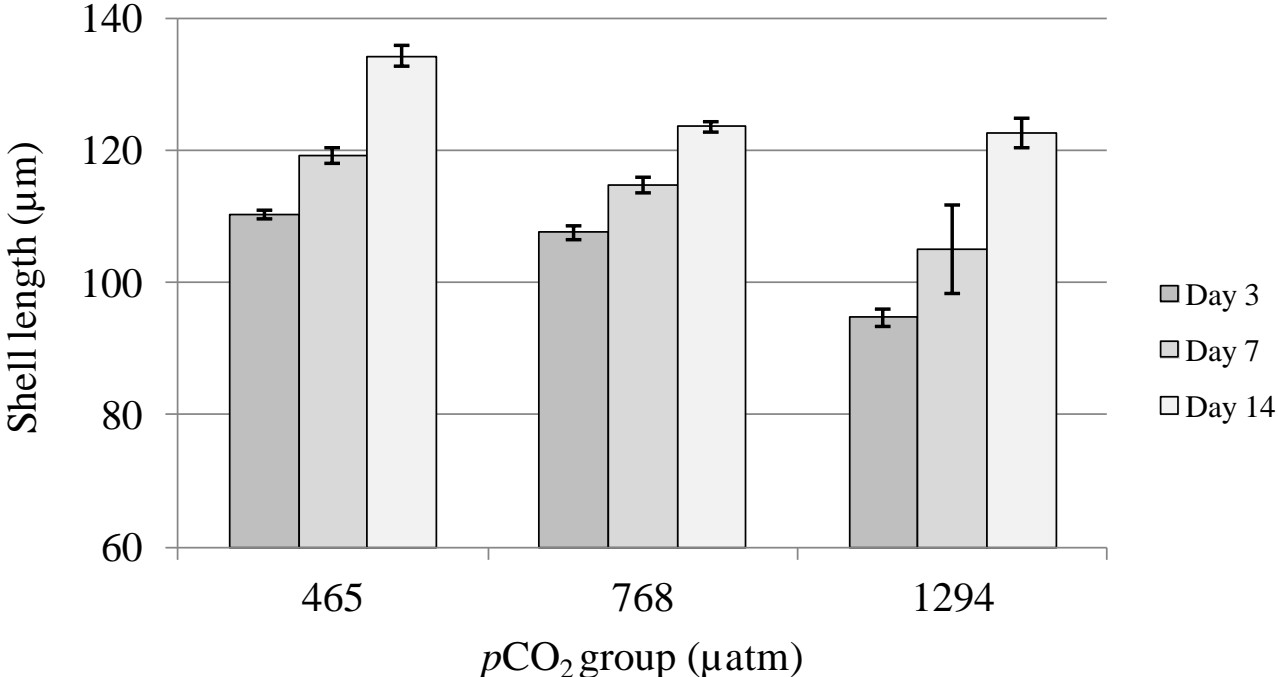

**Figure 5: Shell shape categories.**

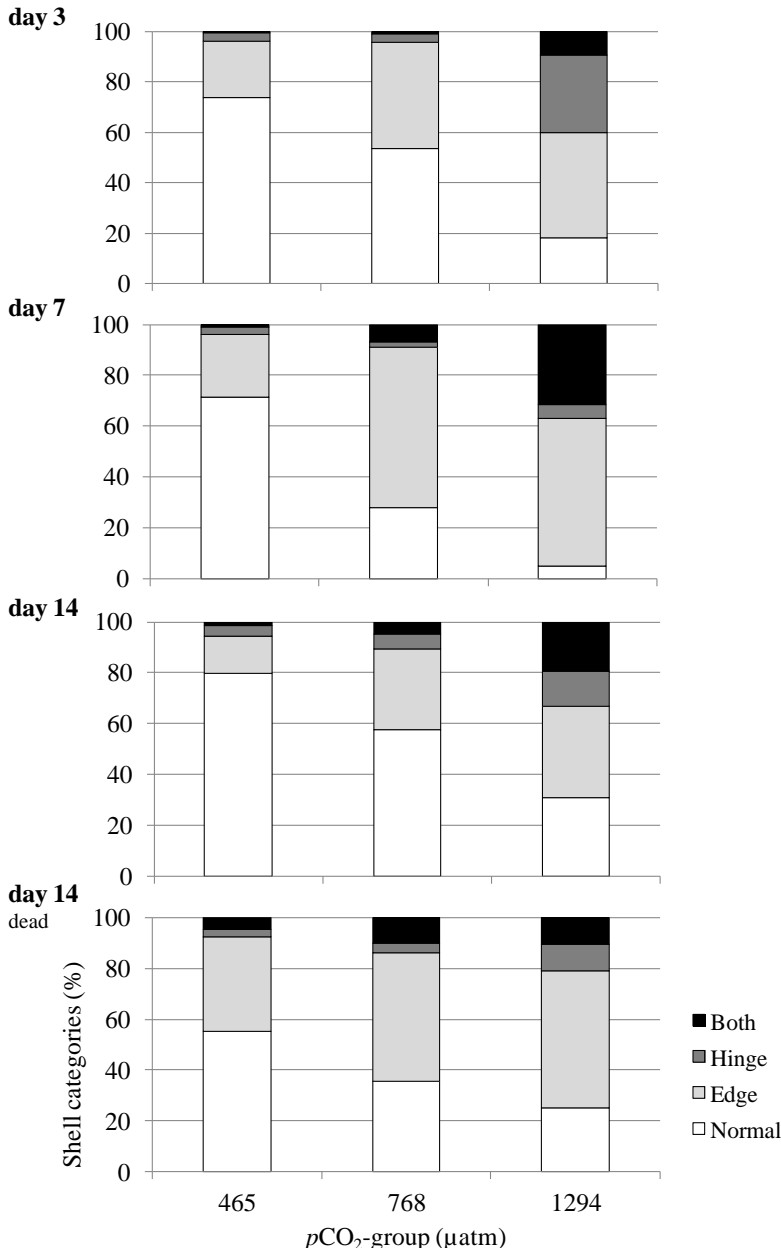