# Peer review of "Sensitivity towards elevated $pCO_2$ in great scallop (*Pecten maximus* Lamarck) embryos and fed larvae."

_Biogeosciences, 2016_

## Referee Comment (RC1) · Anonymous Referee #1 · 7 Jul 2016

1. General comments The paper presents data describing the sensitivity of early-life stages (ELS) of great scallop to future acidification conditions while being well-fed. Results indicate that ELS are particularly sensitive to elevated pCO2, displaying reduced survival, delayed development, and increased abnormalities, and that feeding does not improve this sensitivity. In overall, I commend the authors on a well-written and well-explained paper, whose background rational is clearly explained (a follow-up experiment aimed at deepening their understanding). The data appear robust, and their discussion generally convincing, although there are space for improvements in this part. In my opinion, the discussion could be pushed a bit further, to put the results into a broader context, particularly considering the endpoints considered are not

particularly novel (I am not saying here that these are not useful, on the contrary, but they are the endpoints generally looked at, and the 'so what?' question automatically comes to mind). The authors came to the conclusion that future elevated pCO2 will negatively impact on several aspects of scallops' ELS, despite being well-fed, but do not discuss the implications of their findings. Are there repercussions for aquaculture practices? For population conservation? Therefore, section 4.5 could be improved in order to obtain higher impact. Nevertheless, the paper is throughout really good and deserves to be published. 2. Specific comments In this section, I will list a few remarks and modification that in my opinion could be made to improve the manuscript, of clarify some points, line by line, then comment on the tables and figures. Page 1, in section 1 "Introduction", line 25: "cause elevated CO2 levels" – specify where the CO2 levels are elevated. In the atmosphere? In seawater? That first sentence is a little awkward to read, although still understandable by the reader. Page 2, in section 2 "Materials and Methods": even if it was described in details in the earlier paper (2013), it would be useful to have a brief mention of which seawater parameters were measured and how often, and which were indirectly calculated and how. This would help the understanding of Table 1. Page 3, line 29-31: "At day 3. . . 'protruded velum'". The whole sentence is confusing. Reconsider the grammar (coma?), or rephrase. Do you mean that larvae that have not developed the muscle to retract the velum would be identifiable after being preserved by the presence of a protruded velum? Page 4, section 2.3 "Statistics": I am not the best at commenting on this, but the whole section could be made clearer, from line 15 onwards. E.g. what do you mean by "where Multiple Comparisons were too weak"? Page 5, section 3.2, line 13: "day 3, p<0.000" Is this an error? Page 6, line 4: You mention the coefficient of variation, but you did not mention this earlier in the method section. It would be worth to explain what it is and why you are using it earlier on. Page 7, section 4.3, line 24-25: This last sentence seems like a repeat of what is stated two lines above regarding slower/delayed development– unnecessary, or maybe rephrase saying that (line 21) "The percentages of unshelled larvae at day 2 and larvae with a protruded velum at day 3 were significantly higher in the 1337$\mu$atm

group, which is most likely a result of delayed development caused by elevated pCO2, is in accordance with the reports of slower development at elevated pCO2 levels reported in the earlier study of great scallop larvae (Andersen et al., 2013a) and also in other bivalve larvae (Talmage and Gobler 2011; Kurihara 2008).". In this order in my opinion, it is easier to read and follow the logic, and does not sound like you are repeating yourself. Page 8, line 12-14: so? You are just stating facts, but not trying to say more about it. Page 8, line 19: a reference regarding factors such as genetic variation or energy status would help back-up your explanation. Page 9, line 14: "main bottlenecks in the recruitment process" - reference for this statement? Table 1: Why do you only have AT data for intermediate pCO2 only? How did you calculate the other parameters (CO2SYS? What constants?). One of your aragonite saturation is below 1, how do think this could have affected shell development and growth? Figure 1: The colours used are not consistent between the header tanks and the exposure tanks. Also why are some exposure tanks drawn asides (left/right), and others superimposed (above/below)? Why not all aligned? Figure 5: In my opinion, the graphs would be more easily read if you used 'day 2', 'day 3' ... etc directly on the graph rather than letter A, B, C... But this is just a personal preference. 3. Summary This paper presents interesting results that are in accordance with most of the scientific literature regarding larval development of molluscs under elevated pCO2. It adds valuable insights on the beneficial/neutral effects of added food on the ability of larvae to withstand suboptimal conditions. The paper is clear and generally well-written, and I don't see any major reasons it should not be published, despite minor remarks on my part.

---

## Author Comment (AC1) · 25 Jul 2016

Reply to Anonymous Referee #1

The authors would like to thank Anonymous Referee #1 for useful comments and suggestions. Here are our replies:

1. General comments. AC: We agree that the discussion can be pushed a bit further, especially to look at implications for aquaculture and population conservation, to put the results into a broader context. However, one should be careful not to draw this too far, since experimental conditions can be very different from the natural environment and increase or decrease the sensitivity to a stressor compared to in a natural environment.

[Figure]

2. Specific comments. Page 1, in section 1 - "Introduction", line 25: "cause elevated $CO_2$ levels" – specify where the $CO_2$ levels are elevated. In the atmosphere? In seawater? That first sentence is a little awkward to read, although still understandable by the reader. AC: The first sentence will be changed, since Referee #1 finds it a little awkward.

Page 2, in section 2 "Materials and Methods": even if it was described in details in the earlier paper (2013), it would be useful to have a brief mention of which seawater parameters were measured and how often, and which were indirectly calculated and how. This would help the understanding of Table 1. AC: A brief mention of which seawater parameters were measured and how often, and which were indirectly calculated and how will indeed be included in Material and Methods, page 2 or 3.

Page 3, line 29-31: "At day 3. . . 'protruded velum'". The whole sentence is confusing. Reconsider the grammar (coma?), or rephrase. Do you mean that larvae that have not developed the muscle to retract the velum would be identifiable after being preserved by the presence of a protruded velum? AC: Yes, we mean that larvae that have not developed the muscle to retract the velum would be identifiable after being preserved by the presence of a protruded velum. The sentence will be rephrased.

Page 4, section 2.3 "Statistics": I am not the best at commenting on this, but the whole section could be made clearer, from line 15 onwards. E.g. what do you mean by "where Multiple Comparisons were too weak"? AC:We will try to clarify the whole section, and explain what we mean by "where Multiple Comparisons were too weak".

Page 5, section 3.2, line 13: "day 3, p<0.000" Is this an error? AC: Yes, this is an error. It should read p<0.001

Page 6, line 4: You mention the coefficient of variation, but you did not mention this earlier in the method section. It would be worth to explain what it is and why you are using it earlier on. AC: We agree, and will explain what it is and why we are using it in the method section.

[Figure]
Page 7, section 4.3, line 24-25: This last sentence seems like a repeat of what is stated two lines above regarding slower/delayed development– unnecessary, or maybe rephrase saying that (line 21) "The percentages of unshelled larvae at day 2 and larvae with a protruded velum at day 3 were significantly higher in the 1337$\mu$atm C2 group, which is most likely a result of delayed development caused by elevated pCO2, is in accordance with the reports of slower development at elevated pCO2 levels reported in the earlier study of great scallop larvae (Andersen et al., 2013a) and also in other bivalve larvae (Talmage and Gobler 2011; Kurihara 2008).". In this order in my opinion, it is easier to read and follow the logic, and does not sound like you are repeating yourself. AC: Line 21-25 will be rephrased according to the suggestion made by Anonymous Referee #1

Page 8, line 12-14: so? You are just stating facts, but not trying to say more about it. AC: Line 14 will continue by stating that: The similar shell length at day 7 for the two larval batches suggests that their shell growth rate was similar. Andersen et al. (2013) showed that food availability did not affect larval shell growth the first 6 days after fertilization, supporting that feeding probably did not cause any difference in growth rate between the two larval batches.

Page 8, line 19: a reference regarding factors such as genetic variation or energy status would help back-up your explanation. AC: We will add references as suggested.

Page 9, line 14: "main bottlenecks in the recruitment process" - reference for this statement? AC: Reference will be added.

Table 1: Why do you only have AT data for intermediate pCO2 only? How did you calculate the other parameters (CO2SYS? What constants?). One of your aragonite saturation is below 1, how do think this could have affected shell development and growth? AC: In the table text we explain that ". . .mean salinity and total alkalinity based on two analyses per treatment at the start and the end date, (n=6); ** One mean was used for all groups" But the value can be given in all columns, as for salinity. Also, this

(together with CO2SYS and constants) will be clarified in the additional information that will be given in Material and Methods (as described above for Page 2, in section 2). The aragonite saturation below 1 can possibly add energetic stress to the larvae, since calcium carbonate dissolves at saturation below 1. However, the carbonate shell in live larvae is covered by a protein layer, and the effect may not be significant at 0.82. This will be added to the Discussion using proper references.

Figure 1: The colours used are not consistent between the header tanks and the exposure tanks. Also why are some exposure tanks drawn asides (left/right), and others superimposed (above/below)? Why not all aligned? AC: Colors between the header tanks and the exposure tanks will be corrected, and the exposure tanks all aligned, as there are no reasons why they are not.

Figure 5: In my opinion, the graphs would be more easily read if you used 'day 2', 'day 3' . . . etc directly on the graph rather than letter A, B, C. . . But this is just a personal preference. AC: We agree, and will change Figure 5 accordingly.

---

## Referee Comment (RC2) · F. Gazeau (Referee) · 15 Nov 2016

The manuscript from Andersen and collaborators deals with an important issue related to the impact of ocean acidification on the early larval development of a commercially important species, the great Scallop. This is a concise and well written manuscript that clearly presents the results and provide clear interpretations without over-extrapolating these data based on a controlled laboratory experiment. For all these reasons, I would like to congratulate the authors and will definitely recommend this manuscript for publication in Biogeosciences.

That being said, I have few comments and suggestions I would like the authors to consider before potential acceptance by the editor.

I know the experimental procedure has been published in previous papers but I would strongly recommend the authors to provide more details on 1) how pH was controlled (this appears on Figure 1, but a small paragraph in the text would be useful for the reader), 2) how were pH and total alkalinity measured (I believe these are the 2 "measured" parameters as opposed to pCO2 that is computed) and 3) how did you compute the non measured parameters, CO2-sys? which constants were used? Please also explain how "live" larvae were identified in your formalin preserved samples.

I deeply regret that the authors did not measure and did not report pH on the total scale. For marine environments (their salinity level is 35.1), this is the recommended scale to use. This makes it harder to compare to other studies performed in marine waters as there are no easy way to convert between the 2 scales. Please have a look at the "Best practices guide" that has been released already some years ago: https://www.iaea.org/ocean-acidification/act7/Guide%20best%20practices%20low%20res.pdf. and please consider using this scale for future studies. Anyway, what is done is done, therefore I would strongly recommend to refer whenever possible to their treatments as offsets from the control: i.e. ∼-0.2 and -0.4 pH units, this would be easier for the reader to quickly understand what are the imposed perturbations.

The authors mention that these species is "commercially important" but do not provide justifications for it. Please clarify why this is a commercially important species and provide numbers (yield, income) for it.

As a concluding remark, the authors rightly recommend to conduct future studies considering variable levels of pH/pCO2 as it is the case in many coastal areas, they should further indicate that not only the magnitude of pH variability is potentially different as compared to the open sea but also average levels (especially for these epibenthic species) are certainly far from offshore levels. Just an occasion to insist on the fact that coastal monitoring stations are cruelly missing!

Minor comments: - Please notel that a space is always required between a number and its unit. I will not list them, but many spaces are missing. - P1L27: Not sure this is the right study to cite here, as this is limited to the Arctic. You could cite the meta-analysis of Kroecker et al. (2013) in GCB for instance. - P2L1: Same as above. This will look as self-publicity but you could consider citing Gazeau et al. (2013) in Marine Biology here. - P2L31: correct to: "in 38 L exposure tanks" - P2L33: the authors should mention the levels projected by 2100: -0.06 to -0.32 (by heart, check the right values) according to IPCC 2013. - P6L25: in the standard range of what? - P7L7: survival at day 7 seems higher than what is mentioned here, please check. - Table 1: What is CO2 in ppm ? This is still a partial pressure to me (or a fugacity?) - Figure 1: same comment as Reviewer#1 - Please consider presenting Figure 3 as Figure 2 is presented, i.e. Days in x-axis and pCO2 with different colours.

---

## Author Comment (AC2) · 28 Nov 2016

F. Gazeau: ".....That being said, I have few comments and suggestions I would like the authors to consider before potential acceptance by the editor.

I know the experimental procedure has been published in previous papers but I would strongly recommend the authors to provide more details on how pH was controlled (this appears on Figure 1, but a small paragraph in the text would be useful for the reader) how were pH and total alkalinity measured (I believe these are the 2 "measured" parameters as opposed to pCO2 that is computed) and how did you compute the non measured parameters, CO2-sys? which constants were used? "

Authors: A new section has been added to Material and Methods after review1 (2.1., pg 3 lines 18-31) which describes all the above 1)-3): 2.1. Seawater parameters Seawater was pumped from 160 m depth and filtered through a sand filter before temperature was adjusted in a heat pump. The water was aerated and finally filtered through a 50 $\mu$m filter. Temperature was recorded every 10 minutes using a four detector (one in air and three in exposure tanks) EBI – 1 Ebro 4 temperature logger. The overall mean temperature ($\pm$ SD) calculated from recordings every 10 minutes in three tanks (at the three treatments), was 15.48 $\pm$ 0.16 oC (n=3903). Daily means based on recordings every 10 minutes for each treatment (table 1) was used to calculate pCO2values. Salinity was checked daily using a WTW LF330 Conductivity meter. The pH-level in each exposure tank was measured daily in a 100 ml sample using a Mettler Toledo equipped with a Metler Toledo InLab®ExpertPro pH-probe, calibrated with 4.00 and 7.00 buffers (Certipur® buffer solutions, Merck KGaA, 64271 Damstadt, Germany) traceable to standard reference material from NIST (NBS). The daily means for each treatment (table 1) was used to calculate pCO2 values. Total alkalinity (AT) was analyzed in the three treatments at the start and end of experiment (n=6) by a Titralab, Radiometer, and the mean value 2321.5 $\mu$mol kgSW-1 was used when calculating pCO2values. The pCO2-values ($\mu$atm) corresponding to the pHNIST-values (table 1) were calculated based on the means of temperature (oC), pHNIST, salinity and AT, and using the macro taken directly from Ernie Lewis' "CO2SYS.BAS" Basic Program (Pierrot et al., 2006) with the set of constants K1, K2 from Mehrbach et al. (1973) refit by Dickson and Millero (1987), the constant for KHSO4 from Dickson and Millero (1987) and for total Boron (BT) from Uppstrom (1974). Also, since this section describes the connection between pHNBS and pH NIST, we changed NBS to NIST in the manuscript to avoid confusion. NIST is used in Andersen et al. 2013a.

"Please also explain how "live" larvae were identified in your formalin preserved samples. "

A new sentence has been added on p 5 line 1-2: Larvae were classified as "live"

when the shell was filled with soft tissue, and as "dead" when the shell was empty or contained little soft tissue.

"I deeply regret that the authors did not measure and did not report pH on the total scale. For marine environments (their salinity level is 35.1), this is the recommended scale to use. This makes it harder to compare to other studies performed in marine waters as there are no easy way to convert between the 2 scales. Please have a look at the "Best practices guide" that has been released already some years ago: https://www.iaea.org/oceanacidification/act7/Guideand please consider using this scale for future studies. Anyway, what is done is done, therefore I would strongly recommend to refer whenever possible to their treatments as offsets from the control: i.e. -0.2 and -0.4 pH units, this would be easier for the reader to quickly understand what are the imposed perturbations."

We appreciate the advice and will indeed consider using pH on the total scale in future studies. This work was carried out in 2013 and was focused on comparability with the published work from 2012 (Andersen et al., 2013a). We will refer to offsets from the control (ambient) in both Abstract and Table 1 to show the imposed perturbations.

"The authors mention that these species is "commercially important" but do not provide justifications for it. Please clarify why this is a commercially important species and provide numbers (yield, income) for it."

We mention that this is "a commercially exploited species" (Introduction, p2, line 5-6), and not that it is "commercially important". References to "a commercially exploited species" are added.

"As a concluding remark, the authors rightly recommend to conduct future studies considering variable levels of pH/pCO2 as it is the case in many coastal areas, they should further indicate that not only the magnitude of pH variability is potentially different as compared to the open sea but also average levels (especially for these epibenthic species) are certainly far from offshore levels. Just an occasion to insist on the fact

that coastal monitoring stations are cruelly missing!"

We strongly agree with F. Gazeau to further indicate that not only the magnitude of pH variability is potentially different as compared to the open sea but also that average levels are far from offshore levels. This is included in the Discussion followed by a reference. We also agree that coastal monitoring stations are cruelly missing. We added 3 sentences to the Discussion (now p 11, line 4-9):"Not only are the fluctuations different between open ocean and coastal areas, also average levels in coastal areas are different from open ocean levels. The few reports on the situation in near shore waters show pH values as low as 7.6, already exceeding the expected average values for the open ocean within year 2100 (e.g. Thomsen et al. 2010; Gazeau et al. 2011; Reum et al. 2014). These data are so far based on very few coastal monitoring stations, and effort should be made to increase the monitoring of highly productive coastal areas in the future to reveal the pCO2 levels the coastal epibenthic species in fact are exposed to."

"Minor comments: - Please notel that a space is always required between a number and its unit. I will not list them, but many spaces are missing. "

Many spaces have been added (now p 8 and 9).

"- P1L27: Not sure this is the right study to cite here, as this is limited to the Arctic. You could cite the meta-analysis of Kroecker et al. (2013) in GCB for instance. – P2L1: Same as above. This will look as self-publicity but you could consider citing Gazeau et al. (2013) in Marine Biology here."

We agree with F. Gazeau, and have changed the reference from AMAP (2013) to Kroecker et al. (2013) in GCB. Also, we added Gazeau et al. (2013) in Marine Biology.

"- P2L31: correct to: "in 38 L exposure tanks" "

Corrected.

"- P2L33: the authors should mention the levels projected by 2100: -0.06 to -0.32 (by heart, check the right values) according to IPCC 2013."

Based on IPCC 2013 (FAQ – p 15) we added the sentence: IPCC (2013) has projected the pH levels by 2100 to be 0.06 to 0.32 lower than it is today.

"- P6L25: in the standard range of what?"

We added "...described by (Magnesen et al., (2006) and Andersen et al., (2013b)." (now pg 7, line 30)

"- P7L7: survival at day 7 seems higher than what is mentioned here, please check".

F. Gazeau are right. The numbers for survival were changed due to a correction in sampling volume, but was not changed in the Discussion. The numbers in Discussion are now corrected (now pg 8 line 9 and 13)

"- Table 1: What is $CO_2$ in ppm?"

It is the mole fraction (also called ppmv), but calculated in CO2SYS as part of dry air. It was included due to old reports that used ppm.

"This is still a partial pressure to me (or a fugacity?)" No, it is not fugacity.

"- Figure 1: same comment as Reviewer1"

The figure has been changed accordingly

"- Please consider presenting Figure 3 as Figure 2 is presented, i.e. Days in x-axis and $pCO_2$ with different colours."

Figure 2 does not have $pCO_2$ with different colours, just shades of grey. All measurements for the first group ("Unshelled") was from day 2, and for the second measurements (Protruded velum) from day 3, but not with a direct link between the two (no indication that the unshelled larvae on day 2 ended up with a protruded velum on day 3). We think therefore the original figure 3 is the best way of presenting these data.

[Figure]

---

## Author Response (AR1)

**Response to reviews – a list of relevant changes in the manuscript**

**Reply to Referee#1**

- We agree that the discussion can be pushed a bit further, especially to look at implications for aquaculture and population conservation, to put the results into a broader context. However, one should be careful not to draw this too far, since experimental conditions can be very different from the natural environment and increase or decrease the sensitivity to a stressor compared to in a natural environment. A sentence was included in the section 4.5 (Concluding remarks and future work ), P 10, lines 27-29 in marked up manuscript.

- The first sentence was changed, since Referee #1 finds it a little awkward. "...likely caused elevated $CO_2$ levels, .." was changed to "...likely caused elevated seawater $CO_2$ levels, and thereby lowered the average oceanic pH values,.."

- Seawater parameters were included in a new section in Material and Methods (2.1., pg 3 lines 18-31) which describes in more details how pH was controlled, how pH and total alkalinity were measured, how the non-measured parameters were computed (CO2-sys) and which constants were used.

- Original manuscript Page 3, line 29-31: We mean that larvae that have not developed the muscle to retract the velum would be identifiable after being preserved by the presence of a protruded velum. The sentence was rephrased.

- Original manuscript Page 4, section 2.3 "Statistics" was changed to clarify what we mean by ""where Multiple Comparisons were too weak".

- Original manuscript Page 5, section 3.2, line 13: "day 3, p<0.000" is an error and was changed to p<0.001

- We now explain in Material and Methods what Coefficient of Variation is (p 5 lines 6-9 in marked up manuscript)

- Page 7, section 4.3 in original manuscript, Line 21-25, was rephrased according to the suggestion made by Anonymous Referee #1.

- Page 8, line 12-14 in original manuscript was continued with the sentences: "The similar shell length at day 7 for the two larval batches suggests that their shell growth rate was similar. Andersen et al. (2013) showed that food availability did not affect larval shell growth the first 6 days after fertilization, supporting that feeding probably did not cause any difference in growth rate between the two larval batches."

- Page 8, line 19 in original manuscript (now p 9 line29-30): The sentence  "It is known that the variation in performance between larval batches of P. maximus is very high (Andersen et al.,

2011), and a variation in egg quality have been suggested as an explanation by Robert and Gérard (1999).» was added to include references.

- Page 9, line 14 (p 10 line31 in marked-up ms): The phrase " ..as this is one of the main bottlenecks in the recruitment process." was changed to "..as this may be one of the main bottlenecks in the recruitment process since low survival in this stage is shown in spat production (Andersen et al., 2011)." to include references.

- Table 1: AT data was added, and parameters for CO2SYS was explained in Material and methods, in the new section (as described earlier). Aragonite saturation effect is now mentioned in the Discussion, section 4.4. p 10 line 16-19 (marked-up manuscript).

- Figure 1 colours was changed to be consistent and the tanks aligned.

- Figure 5 was changed according to the suggestions made by referee#1.

**Reply to F. Gazeau (Referee#2)**

- A new section has been added to Material and Methods after review#1, also, since this section describes the connection between $pH_{NBS}$ and $pH_{NIST}$, we changed NBS to NIST in the manuscript to avoid confusion. NIST is used in Andersen et al. 2013a.

- A new sentence has been added on p 5 line 1-2 to explain how "live" larvae were identified in the formalin preserved samples.

- We appreciate the advice to report pH on the total scale, and will indeed consider using pH on the total scale in future studies. This work was carried out in 2013 and was focused on comparability with the published work from 2012 (Andersen et al., 2013a). We will refer to offsets from the control (ambient) in both Abstract and Table 1 to show the imposed perturbations.

- We did not mention that this species is "commercially important", but mentioned that this is "a commercially exploited species» (Introduction, p2, line 5-6). References to "a commercially exploited species» are added.

- We strongly agree with F. Gazeau to further indicate that not only the magnitude of pH variability is potentially different as compared to the open sea but also that average levels are far from offshore levels. This is included in the Discussion followed by a reference. We also agree that coastal monitoring stations are cruelly missing. We added 3 sentences to the Discussion (now p 11, line 4-9).

- Missing spaces have been added (now p 8 and 9).

- We agree with F. Gazeau that AMAP (2013) is not the right study to cite here, as this is limited to the Arctic., and have changed the reference from AMAP (2013) to Kroecker et al. (2013) in GCB. Also, we added Gazeau et al. (2013) in Marine Biology.

- We corrected "- P2L31" to "in 38 L exposure tanks" "

- We mention the pH levels projected by 2100 according to IPCC (2013) to be 0.06 to 0.32 lower than it is today.

- We added "…described by (Magnesen et al., (2006) and Andersen et al., (2013b)." (now pg 7, line 30) to explain "- P6L25: in the standard range of what?"

- The numbers for survival at day 7 in Discussion was corrected (now pg 8 line 9 and 13).

- $CO_2$ in ppm in Table 1 is the mole fraction (also called ppmv), but calculated in CO2SYS as part of dry air. It was included due to old reports that used ppm.
- It is not fugacity.

- Figure 1 has been changed accordingly to referee 1

- Figure 2 does not have $pCO_2$ with different colours, just shades of grey. All measurements for the first group ("Unshelled") was from day 2, and for the second measurements (Protruded velum) from day 3, but not with a direct link between the two (no indication that the unshelled larvae on day 2 ended up with a protruded velum on day 3). We think therefore the original figure 3 is the best way of presenting these data, and do not want to present Figure 3 as Figure 2 is presented, i.e. Days in x-axis and $pCO_2$ with different colours.

[revised manuscript text omitted]